# Unlocking Consumer Preferences: Sensory Descriptors Driving Greek Yogurt Acceptance and Innovation

**DOI:** 10.3390/foods14010130

**Published:** 2025-01-05

**Authors:** Helena Maria Andre Bolini, Flavio Cardello, Alessandra Cazellatto de Medeiros, Howard Moskowitz

**Affiliations:** 1Department of Food Engineering and Technology, University of Campinas, UNICAMP Monteiro Lobato, 80, Campinas 13083-862, SP, Brazil; f263480@dac.unicamp.br (F.C.); acmls@unicamp.br (A.C.d.M.); 2Independent Researcher, White Plains, NY 10604, USA; mjihrm@gmail.com

**Keywords:** Greek yogurt, sensory attributes, consumer preferences, functional food, product optimization, market trends

## Abstract

Greek yogurt, a traditional food with roots in Ancient Greece, Mesopotamia, and Central Asia, has become a dietary staple worldwide due to its creamy texture, distinct flavor, and rich nutritional profile. The contemporary emphasis on health and wellness has elevated Greek yogurt as a functional food, recognized for its high protein content and bioavailable probiotics that support overall health. This study investigates the sensory attributes evaluated by a panel of 22 trained assessors and the consumer preferences driving the acceptance of Greek yogurt formulations. Samples with higher consumer acceptance were characterized by sensory attributes such as “high texture in the mouth, surface uniformity, creaminess, apparent homogeneity, mouth-filling, grip in the mouth, ease of pick-up with a spoon, milk cream flavor, sweetness, and dairy flavor” (Tukey’s test, *p* < 0.05). These attributes strongly correlated with consumer preferences, underscoring their importance in product optimization. The findings provide a framework for refining Greek yogurt formulations to address diverse market demands, achieving a balance between sensory excellence and practical formulation strategies. This research reinforces the significance of Greek yogurt as a culturally adaptable, health-promoting dietary component and a promising market segment for ongoing innovation.

## 1. Introduction

Yogurt, a food with ancient origins, has been integral to diets across civilizations such as Ancient Greece, Mesopotamia, and Central Asia. Its production began with the spontaneous fermentation of milk in animal-skin containers, resulting in a creamy texture and distinct flavor [1]. Over time, yogurt has been woven into cultural traditions worldwide, becoming a staple in many regions.

Among its many variations, Greek yogurt has recently gained global acclaim for its thick texture and slightly acidic taste, along with notable nutritional benefits [2,3,4]. The straining process used to make Greek yogurt removes whey, yielding a product with higher protein content and lower levels of lactose and carbohydrates [5,6]. This process concentrates essential nutrients, such as calcium, probiotics, and vitamins, while contributing to the yogurt’s creamy consistency and tangy flavor—attributes that have heightened consumer appreciation [7].

Notably, Greek yogurt contains up to twice the protein of traditional yogurt, further enhancing its reputation as a nutritious and versatile food option [8,9].

The contemporary emphasis on health and wellness has driven the promotion of Greek yogurt as a functional food, rich in bioavailable proteins and probiotics that support digestion, bone health, and muscle function [9,10]. Greek yogurt is a source of calcium, phosphorus, potassium, zinc, and A and B12 vitamins [11].

Its popularity is evident in market trends, with Greek yogurt representing a significant share of global sales, especially in regions like North America and Europe, where demand for health-oriented, convenient foods is high [12].

Despite its success, producing low-fat Greek yogurt presents notable challenges. Reducing fat content adversely affects sensory properties, particularly texture and flavor [13].

As a result, research has explored the use of whey protein concentrates and hydrocolloids to mitigate these sensory deficits [5,12]. Innovations in formulation are essential to maintaining product appeal and nutritional integrity, addressing evolving consumer preferences, and supporting industry growth [14].

Understanding consumer preferences and sensory attributes is pivotal for advancing Greek yogurt products. Sensory studies have identified key attributes, such as texture and flavor profile, that influence consumer satisfaction and drive product development [15,16]. These evaluations are critical for optimizing production processes and ensuring consistent product quality, aligning with consumer expectations [16,17].

In summary, Greek yogurt stands out as a highly desirable functional food due to its rich nutritional and sensory profile [18]. Ongoing research and innovation, underpinned by consumer insights, remain vital for the continued expansion and success of this market segment. Furthermore, yogurt’s widespread appeal and adaptability underscore its cultural significance and enduring role as a health-promoting dietary component [19].

From its artisanal roots to large-scale industrial production beginning in the early 20th century, yogurt has evolved to meet modern demands, providing an accessible, nutritious, and enjoyable food choice [20].

Key to yogurt’s sensory appeal is its flavor compounds, derived from milkfat lipolysis and microbial transformations of lactose and citrate [17]. Over 100 volatile compounds contribute to yogurt’s characteristic aroma and taste, with lactic acid, acetaldehyde, diacetyl, and acetoin among the primary contributors [7]. Understanding these complexities continues to shape innovations in yogurt production, ensuring products remain flavorful and nutritionally robust.

Considering the importance of determining the descriptive sensory profile and consumer acceptance, since they are factors that significantly influence the sensory quality of foods [21], this study aimed to identify the sensory descriptors that positively drive consumer preference for Greek yogurt. The findings are intended to contribute to innovation and the development of new products that meet market demands. The application of sensory evaluation methods serves as a valuable tool for optimizing production processes and ensuring consistent product quality for consumers [22].

In summary, Greek yogurt, due to its nutritional and sensory characteristics, stands out as a highly demanded functional food. Continuous development and innovation, grounded in understanding consumer preferences, are essential for the industry’s success and for the expansion of the global yogurt market.

## 2. Material and Methods

### 2.1. Samples

Eleven traditional Greek yogurt samples were analyzed, including three Brazilian market leaders, two produced by multinational companies (coded 1-GYMML and 2-GYMML), and one produced by national company (coded 3-GYNML). Another eight companies that developed Greek yogurts, were prepared in the LCSEC (Prototype 1 to 8), and presented variations in sugar syrup, cream milk, milk protein concentrate, concentrated milk fat, culture for yogurt, and presence or no presence of modified starch and tasteless and colorless gelatine, were also analyzed. The basic formulation of the eight prototypes (P1; P2; P3; P4; P5; P6; P7; P8) are presented in Table 1.

### 2.2. Methods

#### Sensory Analysis

The three commercial samples were obtained from local supermarkets in Campinas (São Paulo State, Brazil). Approximately 30 g of each sample, refrigerated at 4 °C, were served in 30 mL disposable cups coded with random three-digit numbers for sensory analysis. Water and cream-cracker biscuits were also provided for palate cleansing. The study was approved by the Ethics Committee of the State University of Campinas (91178118.0.0000.5404), and informed consent was obtained from all volunteers.

Sensory analyses were conducted at the Laboratory of Sensory Science and Consumer Studies (LCSEC) of the School of Food Engineering (FEA), University of Campinas (UNICAMP), according to official methods, American Society of Testing and Materials, consumer study [23], and ISO 8589:2007 standards [24]. The sensory panelists included young adults and adults (aged 18 to 40 years) who were regular Greek yogurt consumers. Participants were recruited through advertisements posted on flyers and social media platforms. For the experimental design, all sensory tests utilized a complete block design with balanced sample presentation [25].

Prior to product evaluation, all panelists received and signed an Informed Consent Form as required by the UNICAMP Research Ethics Committee. This document, which outlined the research details, was presented to ensure full understanding and to confirm participants’ willingness to partake in the study.

### 2.3. Descriptive Quantitative Analysis

#### Pre-Selection of Assessors

To efficiently analyze the data of descriptive sensory profiles, the panel members must have discriminatory power [26]. Thus, a preselection applied with triangular tests was conducted of the pre-candidate assessors for the Quantitative Descriptive Analysis^®^ [27] of Greek yogurt. A total of 25 pre-candidates were recruited, all consumers of the product having no restrictions in consuming the product and showing interest in taking part in the test.

The candidates were instructed to evaluate the samples from left to right and identify which coded sample was different from the others. The discriminative power was evaluated by Wald’s sequential analysis [28], using triangular difference tests with a significant difference at the 1% level with respect to sweetness, the objective being to select the candidates best able to discriminate the samples.

Each candidate performed three triangular tests per day to preserve their sensory capacity. The parameters used to analyze the discriminatory capacity in Wald’s sequential analysis were prefixed at ρ0 = 0.45 (maximum acceptable lack of ability) and ρ1 = 0.70 (minimum acceptable ability), as well as for the risks α = 0.05 (probability of accepting a candidate without acuity) and β = 0.05 (probability of rejecting a candidate without acuity) [29]. Having defined the parameters, two lines of equation were obtained and used to construct the Wald graph, while three defined areas were obtained: acceptance, indecisive, and rejection area of the panelists [29].

Thus, the panelists were selected or rejected according to the number of correct replies in the triangular tests applied and projected in the Wald graph [20]. In the end, 22 panelists were selected for the QDA^®^ according to ISO 8586 [27] of Greek yogurt samples.

### 2.4. Quantitative Descriptive Analysis

Twenty-two selected panelists (12 women and 10 men, mean age = 32) were recruited based on their willingness to participate and their consumption of Greek Yogurt on a regular basis. Six 1 h training sessions were conducted. The samples that were evaluated were gently stirred to homogenize them, filled in 30 g plastic containers, covered, and stored in the refrigerator at 4 °C until served to the assessors. The samples were coded with three-digit random numbers.

Table 2 summarizes the descriptor terms, along with definitions, anchor words, and references to none/weak (0.0), moderate/medium (4.5), and strong (9.0), which are used for training and selection of assessor panels.

The participants selected presented *p*-value of sample <0.50 and *p*-value of repetition >0.05 and had agreement among them, respectively, for the 41 descriptor terms [30]. The intensity for each descriptor term was marked on a nine-point scale with Compusense^®^ software (V. 24.0.26998, Compusense Inc., Guelph, ON, Canada). The assessors (highly trained individuals) were the analysis in triplicate. Significant means for sensory analyses were separated by Tukey’s honestly significant difference. Significance was pre-established at alpha < 0.05. Principal component analysis was performed using the means obtained from descriptive analysis (QDA^®^) [30].

The potential first-order carry-over effect was mitigated through a balanced design based on MacFie et al. (1989) [25]. The data were collected using Compusense^®^ software (V. 24.0.26998, Compusense Inc., Guelph, ON, Canada).

Statistical analysis of analysis of variance, Tukey’s test, and multivariate principal component analysis to check the preference mapping was performed using SAS^®^ software (V.9.4, 2024, SAS, Cary, NC, USA).

### 2.5. Acceptance Analysis

An acceptance [30] test was conducted with 150 frequent consumers of plain Greek yogurt (67 men and 83 women, mean age = 33 years). Samples were served over 2 consecutive days. Consumers were asked to mark their liking in relation to appearance, flavor, texture, and overall acceptability using the nine-point hedonic scale (from 1 = dislike extremely to 9 = like extremely).

The inclusion criteria required participants to consume yogurt at least once a week and to have no milk allergies. The tests were conducted in a controlled environment (temperature maintained at 20 °C), where 30 mg of each sample were served at 5 °C in plastic cups labeled with three-digit codes. Prior to participation, all individuals read and signed an informed consent form. To cleanse their palate between samples, participants were provided with cream-cracker biscuits and water.

### 2.6. Physical–Chemical Tests

Physicochemical analyses (pH determination, soluble solids content, and instrumental texture analysis) were conducted at the Central Instrumental Laboratory of the School of Food Engineering, State University of Campinas (UNICAMP).

#### 2.6.1. pH

The pH of the yogurt samples was determined using an electrometric method with a pH meter in an Orion Expandable Ion Analyzer EA 940 [31]. The measurements were taken after the package was opened. A 20-minute interval was chosen between each evaluation for each measurement. The analyses were performed in triplicate. The results were analyzed using the SAS program through analysis of variance and Tukey’s mean tests (*p* < 0.05).

#### 2.6.2. Soluble Solids

The concentration of soluble solids was determined with direct reading in a Carl ZEISS Jena bench refractometer, according to method no. 932.12 of the AOAC [31]. It was performed in triplicate at a temperature of 20 °C, and the results were expressed in °Brix.

#### 2.6.3. Instrumental Texture

Texture analysis was performed following the method of Rawson and Marshall (1997) [32] using a TAXT2 universal texture analyzer equipped with a 35 mm diameter flat-bottom cylindrical probe (A/BE 35). The results were processed using Texture Expert software version 1.11 for Texture Profile Analysis (TPA).

### 2.7. Statistical Analysis of Data

Data from quantitative descriptive analysis (QDA), consumer tests, and physicochemical tests were analyzed using univariate analysis of variance (ANOVA) and Tukey’s mean comparison tests. The mean results from descriptive terms, consumer tests, and physicochemical analyses were further analyzed using multivariate statistical analysis, specifically partial least-squares regression [33].

## 3. Results

The results presented in this study can serve as a valuable guideline for the formulation of developing Greek yogurts, as well as for existing products on the market, by highlighting sensory characteristics of importance to consumers.

For instance, samples P1, P3, P5, P8, and 1-GYMML could utilize the findings from the present study to enhance dairy flavor (milk), milk cream flavor, and texture in the mouth while simultaneously reducing metallic flavor, cheese flavor (sour milk), sulfurous flavor, and rancid flavor to improve product quality and acceptance.

It is important to emphasize that these adjustments should also be supported by additional information about the products, as well as manufacturing and storage conditions, such as shelf life, environmental factors, ingredient temperatures, and other relevant data to better inform decision-making.

Figure 1 represents the results obtained from hierarchical cluster analysis, showing the formation of four similarity groups based on the descriptive sensory profile.

The first group comprised samples P4, P7, 2-GYMML, and 3-GYNML, characterized by high intensities of dairy aroma, sweet aroma, texture in the mouth, surface uniformity, creamy in the mouth, apparent uniformity, mouth-filling, sweetness, milk cream flavor, and dairy flavor. These samples also exhibited low or negligible intensities of rancid oil flavor, metallic, sulfurous, salty, lump formation, grip in the mouth, syneresis, bitterness, cheese flavor, and cottage-cheese flavor. Additionally, they presented intermediate levels of acidic aroma, acidity, and viscosity.

The second group included yogurt samples P2, P6, and P8, which were characterized by low intensities of metallic flavor, cottage-cheese flavor, lump formation, sulfurous cheese flavor, and cheese flavor. Conversely, these samples showed high intensities of texture in the mouth, surface uniformity, homogeneity and uniformity, and intermediate levels of dairy flavor and milk cream flavor.

The third group consisted of yogurt samples P1, P3, and P5, primarily marked by low intensities of cottage-cheese flavor, lump formation, and bitterness. They exhibited high intensities of dairy aroma, sweet aroma, texture in the mouth, creamy in the mouth, apparent uniformity, and mouth filling, along with intermediate levels of caramel flavor and acidity.

The fourth group was represented by a single sample, 1-GYMML (the least preferred), which showed low intensities of buttery flavor, diacetyl aroma, vanilla flavor, and caramel aroma. However, it was characterized by high intensities of filament formation when picked up with a spoon, viscous texture, bitterness, cheese flavor, and residual bitterness. It also presented intermediate levels of dairy aroma, sweet aroma, texture in the mouth, surface uniformity, mouth-filling, buttery aroma, and caramel flavor.

Samples with higher acceptance formulations (2-GYMML, 3-GYNML, P4, and P7) exhibited characteristics like “high texture in the mouth, surface uniformity, creaminess in the mouth, apparent homogeneity, mouth-filling, grip in the mouth, texture when picked up with a spoon, milk cream flavor, sweetness, dairy flavor, and higher sweetness”, as shown in Table 2 and evidenced in Figure 1. These attributes are significantly higher compared to the other samples (*p* < 0.05), providing valuable insights for optimizing yogurt formulations. 

In Figure 2, it is possible to observe the importance of flavor in the overall impression of the product, as the acceptance results for these two variables exhibit the same trend.

It is important to highlight that the preference drivers (Figure 3 and Figure 4) identified in this study were defined based on a consumer group predominantly composed of women, aged 18–35 years, and self-employed or university-employed individuals. When considering the information from the results of the descriptive sensory profile and the overall consumer acceptance presented, it is essential to consider the characteristics that contribute positively or negatively, as well as those that do not influence consumer preference, as evidenced in Figure 3, which presents the acceptance ratings of individuals on the hedonic scale, and Figure 4, which illustrates the mean scores provided by consumers for an overall impression. However, it is crucial to consider the minimum and maximum limits of the ingredients used in each formulation to achieve a successful response.

It is also evident that even high acceptance, appearance, consistency in the spoon, and texture do not correspond to the overall impression.

This finding is supported by the results in Table 3, where the samples with the mentioned characteristics demonstrated higher acceptance regarding flavor and overall impression (Figure 2). Furthermore, Figure 5 highlights that these same samples received a higher proportion of responses, indicating “I would certainly buy it”. Table 4 provides demographic information about the participants, revealing that the majority of Greek yogurt consumers in the study were women. Most participants held a bachelor’s degree, were aged between 31 and 35 years, and were employed in academia or were self-employed.

Flavor appears to be the most critical factor influencing consumers’ purchasing decisions, as the trends observed for liking of overall impression and liking of flavor are closely aligned. Additionally, consumers exhibited a preference for samples with pronounced creamy milk flavor characteristics, coupled with a smooth and creamy mouthfeel.

The preference did not differ significantly between 2-GYMML, 3-GYNML, P4, and P7 (*p* > 0.05). This is a relevant result, as it demonstrates the development of two formulations with acceptance levels comparable to the market-leading Greek yogurt. Greek yogurt represents one of the most rapidly expanding categories within the dairy sector. Also known as strained yogurt, it is produced by removing whey through a straining process, resulting in a product with higher total solids and reduced lactose content compared to conventional yogurt. Due to its concentrated nature, Greek yogurt exhibits distinctive sensory properties [34].

The results, presented in Table 5, highlight the limits found in the four preferred samples of Greek yogurt, relative to solid soluble with values 18.00 to 18.86 °Brix (no significant difference *p* > 0.05), pH values 4.17 to 4.26 (no significant difference *p* > 0.05), and texture from 18.264 to 30.900 F/g (means significantly different at *p* < 0.05).

A study conducted by Pappa et al. [35] mentions that a pH range of 4.1–4.6 is preferable for developing the desired flavor, which can be considered the main characteristic of Greek yogurt—exactly the range found in the samples in Table 3. Additionally, this range is important for the aggregation of casein particles, the formation of a coagulum avoiding syneresis, and the prevention of the growth of undesirable microorganisms [28,29,30].

The solid soluble (°Brix) values found in the present research were in line with those reported in the literature with Greek yogurt [36,37,38].

However, it is important to highlight that the results presented in this article are limited to Greek yogurt and the sensory evaluation methodologies discussed. The insights provided aim to establish a scientific foundation for the Greek yogurt industry, enabling the optimization of existing products and fostering the development of innovative formulations through the application of food sensory analysis techniques.

Brow and Chambers [39] studied the descriptive sensory profile of commercial Greek yogurt samples and two prototypes produced in a local laboratory (Kansas University). The prototypes, which required less processing input and could, therefore, be considered “more sustainable” than current products, closely resembled both the flavor and texture of market-leading Greek yogurts. The authors noted the potential viability of the lab-made Greek yogurts. The results, observed by Brow and Chambers, align with those obtained in the present study.

The results presented in Figure 1 (highlighting clusters formed based on similarities in descriptive sensory terms), along with the findings from Figure 3 (external preference mapping obtained using the descriptive sensory profile and consumers’ individual ratings of overall impression) and Figure 4 (partial least-squares standardized coefficients of Greek yogurt attributes that positively or negatively influence consumer perception, based on the average overall impression scores), provide insights into the likely reasons behind the mean acceptance ratings for flavor and overall impression shown in Figure 2. These findings also relate to the purchase intention data illustrated in Figure 5, reflecting consumer preferences for the analyzed yogurts in this study.

These results would be very useful information for helping the Greek yogurt industry to improve its current products, as well as develop innovative products for the future [5]. This present study is in line with other published reports [40] related to the textural attributes and the water-holding capacity, which define yogurt quality and determine consumer acceptance. It is important to highlight that these driver preferences were found to consumers aged between 18 and 35 years old.

## 4. Conclusions

The results obtained provide valuable insights into the drivers of preference for Greek yogurt formulations. Samples 2-GYMML, 3-GYNML, P4, and P7 consistently exhibited higher acceptance levels, characterized by attributes such as “high texture in the mouth, surface uniformity, creaminess, apparent homogeneity, mouth-filling, grip in the mouth, ease of pick-up with a spoon, milk cream flavor, sweetness, and dairy flavor”. These sensory attributes showed strong correlations with consumer preferences, emphasizing their relevance in product optimization.

Additionally, the external preference mapping and preference driver analysis validated the importance of specific sensory properties as key determinants of consumer liking. Understanding these attributes enables targeted adjustments to formulations, ensuring optimal sensory appeal while respecting ingredient constraints and production practicality.

This study highlights the critical role of aligning sensory attributes with consumer expectations to improve product acceptance. By exploring sensory characteristics alongside consumer segmentation, the results pave the way for further refinement of Greek yogurt formulations to address diverse market demands. These findings offer a robust framework for designing successful products that achieve a balance between sensory excellence and practical formulation strategies.

## Figures and Tables

**Figure 1 foods-14-00130-f001:**
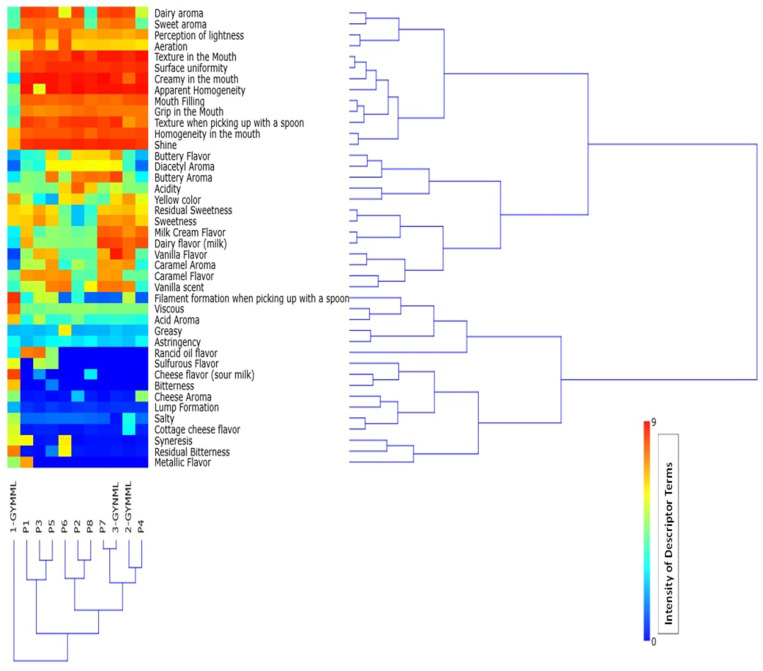
Dendrogram two-way (descriptive sensory terms and yogurt samples) obtained through hierarchical cluster analysis using the Ward method algorithm and the Euclidean distance similarity index. Representative clusters of descriptive terms to Greek yogurt.

**Figure 2 foods-14-00130-f002:**
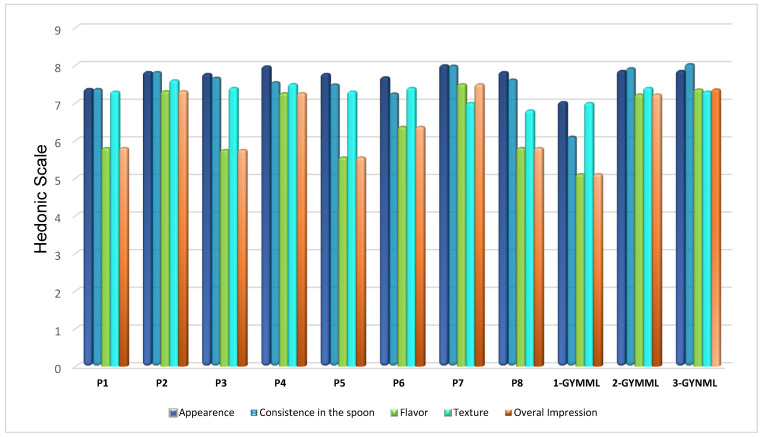
Acceptance of Greek yogurts about appearance, consistency in the spoon, flavor, texture, and overall impression.

**Figure 3 foods-14-00130-f003:**
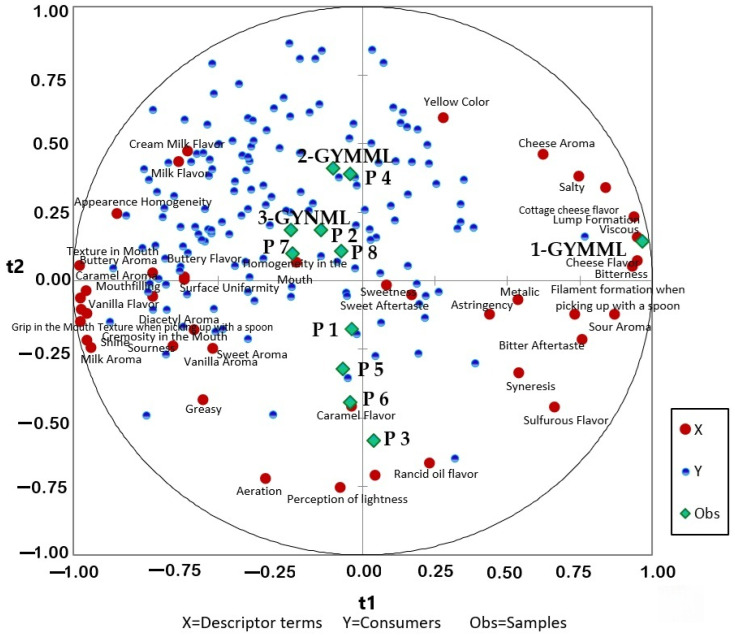
External preference mapping was obtained by partial least-squares regression of the descriptive sensory profile and consumers’ overall impressions of Greek yogurt. (diamond = Greek yogurt samples; blue points = consumers; red points = quantitative descriptive analysis attributes). The partial least-square regression used the individual notes of consumers for overall impression.

**Figure 4 foods-14-00130-f004:**
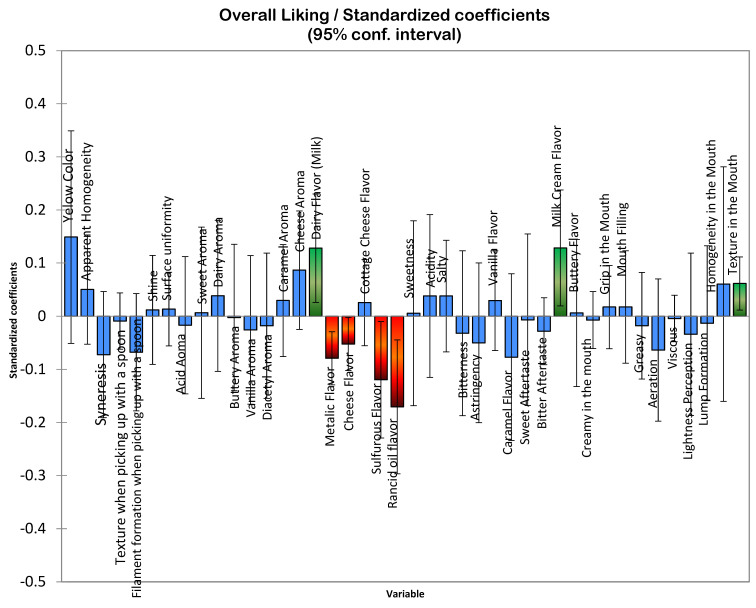
Partial least-squares standardized coefficients of Greek yogurt (green = descriptor terms that contribute positively to consumer acceptance; blue = descriptive terms that did not significantly contribute to consumer acceptance; red = descriptor terms that contribute negatively to consumer acceptance) at 95% confidence interval. The partial least-square regression used the average consumer notes for overall impression.

**Figure 5 foods-14-00130-f005:**
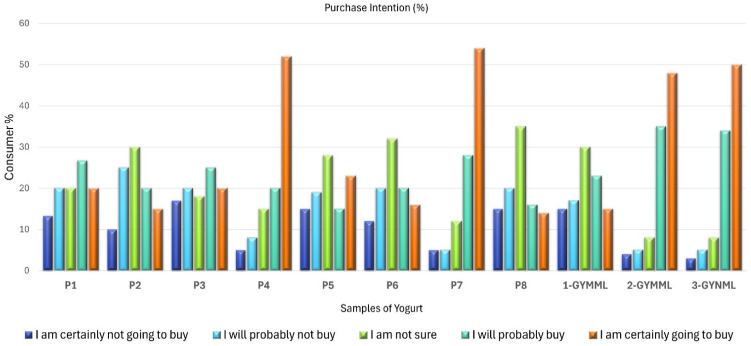
Consumer purchase intention regarding the Greek yogurt samples.

**Table 1 foods-14-00130-t001:** Basic ingredients to formulate Greek yogurt produced in laboratory. The quantities are expressed in 100 g.

Ingredients	P1	P2	P3	P4	P5	P6	P7	P8
Sugar syrup	15.0000	15.0000	15.0000	13.0000	15.0000	13.0000	15.0000	13.0000
Skimmed milk powder	1.4000	1.4000	1.4000	1.4000	1.4000	1.4000	1.4000	1.4000
Cream milk	11.2000	21.5000	13.2000	13.8000	13.2000	13.8000	13.2000	13.8000
Milk Protein Concentrate 70	3.7000	4.8000	3.7000	3.7000	3.7000	3.7000	4.0000	3.7000
Concentrated milk fat: 3.0%/SNF 8.5%	68.6983	57.2983	65.8983	67.7983	65.8983	67.7983	65.9000	67.7983
Culture Yoflex Premium 3.0	0.0017	0.0017	0.0017	0.0017	0.0017	0.0017	0.0017	0.0017
Modifed starch	0.0000	0.0000	0.6000	0.3000	0.6000	0.3000	0.6000	0.3000
Tasteless and colorless gelatin	0.0000	0.0000	0.2000	0.2000	0.2000	0.2000	0.2000	0.2000

**Table 2 foods-14-00130-t002:** Descriptor terms of Greek yogurt.

Descriptor Term	Definition	References
Yellow color	Light yellow color, like butter	None: Pasteurized fresh milkcCream “Fazenda Bela Vista” brand Moderate/Medium: 100 mL of pasteurized fresh milk cream “Fazenda Bela Vista ^®^” brand and 0.25 mL of prepared Oetker^®^ pineapple gelatin, as per package instructions, without allowing it to set Strong: 200 mL of pasteurized fresh milk cream “Fazenda Bela Vista”^®^ brand and 1 mL of prepared Oetker^®^ pineapple gelatin, as per package instructions, without allowing it to set
Homogeneous appearance	Homogeneous, smooth, and lump-free appearance of the surface of the yogurt	Slight/Weak: 100 mL of Salute^®^ brand plain liquid yogurt and 50 g of Nestlé ^®^ cream without whey Moderate/Medium: 100 mL of Salute^®^ brand plain liquid yogurt and 50 g of Nestlé^®^ cream without whey slowly mixed with spoon on a glass plate for 10 s Strong: 100 g of Salute brand plain liquid yogurt and 100 g of Nestlé^®^ cream without whey, blended
Syneresis	Separation of whey from yogurt, accompanied by a reduction in its volume and intensified by changes in temperature, pH, and mechanical factors	None: Greek yogurt Vigor^®^ brand Moderate/Medium: Commercial lactose-free plain yogurt with skimmed milk Strong: Commercial whole plain yogurt containing 10% added water
Texture when picked up with a spoon	Force applied to place the spoon and remove a little of the product to be consumed	Slight/Weak: 100 g of pasteurized fresh milk cream “Fazenda Bela Vista”^®^ brand and 100 g of Salute ^®^ brand plain liquid yogurt Moderate/Medium: 100 g of milk cream Nestlé^®^ Tetrapack package Strong: Canned Desert “Brigadeiro” Nestlé^®^
Filament formation when picked up with a spoon	Filaments formed between the product in the jar and the product in the spoon	None: Pasteurized fresh milk cream “Fazenda Bela Vista” brand Moderate/Medium: 100 mL of pasteurized fresh milk cream “Fazenda Bela Vista^®^” brand and 50 mL of Nestlé^®^ condensed milk Strong: Nestlé^®^ Condensed milk
Shine	Reflection of light on the surface of the product	Slight/Weak: None: Pasteurized fresh milk cream “Fazenda Bela Vista^®^” brand Moderate/Medium: Flan Danette white chocolate Strong: Orange blossom honey Melbee^®^ brand
Surface uniformity	Regularity of the apparent texture on the surface of the product	Slight/Weak: Creme de leite fresco pasteurizado marca Fazenda Bella Vista^®^ Moderate/Medium: Yogurt Grego marca Danone^®^ Strong: 5 mg of butyric acid and 10 mg of caprylic acid in 1 liter of pasteurized fresh cream, Fazenda Bella Vista^®^ brand
Acid aroma	Aroma that is felt when inhaling near a product that contains acids, especially lactic acid	None: Pasteurized fresh cream, Fazenda Bella Vista^®^ brand Moderate/Medium: 100 g of creme de leite fresco pasteurizado marca Fazenda Bella Vista^®^ and 1 mL of lactic acid Strong: 10 g of pasteurized fresh cream, Fazenda Bella Vista^®^ brand, 1 mL of lactic acid, and 50 g of whipped yogurt, Salute^®^ brand
Sweet aroma	Aroma that is felt when inhaling near sweet foods	None: Pasteurized fresh cream, Fazenda Bella Vista^®^ brand Moderate/Medium: Pasteurized fresh cream, Fazenda Bella Vista^®^ brand and 10 mL of Karo Corn^®^ glucose Strong: Homogenized 100 g of pasteurized fresh cream, Fazenda Bella Vista^®^ brand, 20 mL of Karo Corn^®^ glucose, and 20 g of dulce de leche Itambé^®^ brand
Milky aroma	Aroma that is felt when inhaling dairy products	None: Distilled water Moderate/Medium: Whole milk Sheffa^®^ brand Strong: Homogenized 50 g of pasteurized fresh cream, Fazenda Bella Vista^®^ brand, and 100 g of Ninho^®^ powder milk
Vanilla aroma	Aroma that is felt when inhaling near vanilla essence or products containing vanilla	None: Whole milk Sheffa^®^ brand Moderate/Medium: 0.5 mL of IFF^®^ vanilla essence F in 50 mL of whole milk Sheffa^®^ brand Strong: 1 IFF^®^ vanilla essence F in 50 mL of whole milk Sheffa^®^ brand
Aroma amanteigado (diacetil)	Aroma that is felt when inhaling near butter and buttery products (Diacetil)	None: Whole milk Sheffa^®^ brand Moderate/Medium: 50 mL of whole milk Sheffa^®^ brand and 0.0001 mL of diacetyl Sigma Aldrich^®^ PA Strong: 50 mL of whole milk Sheffa^®^ brand and 0.0025 mL of diacetyl Sigma Aldrich^®^ PA
Caramel aroma	Aroma that is felt when inhaling near foods that contain caramelized sugar	None: Ninho^®^ whole milk Moderate/Medium: 20 g of sucrose União^®^ caramelized in 1 liter of Ninho^®^ whole milk UHT Strong: 50 g of sucrose União^®^ caramelized in 1 liter of Ninho^®^ whole milk UHT
Cheese aroma	Aroma that is felt when inhaling near curdled (sour) milk	None: 100 g of pasteurized fresh milk cream “Fazenda Bela Vista^®^” brand Moderate/Medium: 2 mg of butanoic acid and 2 mg of caplilic acid in 100 g of pasteurized fresh milk cream “Fazenda Bela Vista” brand Strong: 5 mg of butanoic acid and 5 mg of caplilic acid in 100 g of pasteurized fresh milk cream “Fazenda Bela Vista^®^” brand
Mikky flavor	Characteristic flavor of instant whole milk powder dissolved in water	Slight/Weak: 50 g of Ninho Nestlé^®^ instant whole milk powder in 1 liter of deionized water Moderate/Medium: 150 g of *Ninho* Nestlé^®^ instant whole milk powder in 1 liter of deionized water Strong: 500 g of Ninho Nestlé^®^ instant whole milk powder in 1 liter of deionized water
Metallic taste	Characteristic flavor of foods containing iron or some canned foods	None: Pasteurized fresh milk cream Verde Campo^®^ brand Moderate/Medium: 50 mL of pasteurized fresh milk cream Verde Campo^®^ brand and 0.0005 g of FeSO_4_·7H_2_O in 10 mL of whole milk Shefa^®^ brand Strong: 50 mL of pasteurized fresh milk cream Verde Campo^®^ brand and 0.001 g of FeSO_4_·7H_2_O in 10 mL of whole milk Shefa^®^ brand
Cheese flavor (sour milk)	Sour milk flavor	None: Pasteurized fresh milk cream Verde Campo brand Moderate/Medium: 50 mL of pasteurized fresh milk cream Verde Campo^®^ brand and 0.25 mg of butiric acid and 0.25 mg of caprilic acid diluted in 10 mL whole milk Shefa^®^ brand Strong: 50 mL of pasteurized fresh milk cream Verde Campo^®^ brand, 0.75 mg of butiric acid, and 0.75 mg of caprilic acid diluted in 10 mL of whole milk Shefa^®^ brand
Cottage cheese flavor	Characteristic flavor of cream cheese	None: Deionized water Moderate/Medium: 200 mL of whole milk Shefa^®^ with 30 g of cream cheese Danúbio^®^ brand Strong: 100 g of cream cheese Danúbio^®^ brand and 100 g of pasteurized fresh milk cream “Fazenda Bela Vista^®^” brand
Sulfurous flavor	Characteristic flavor of freshly peeled boiled egg white	None: Pasteurized fresh milk cream “Fazenda Bela Vista^®^” brand Moderate/Medium: 2 mg of dimethyl sulfide Sigma Aldrich PA in 1 liter of pasteurized fresh milk cream “Fazenda Bela Vista^®^” brand Strong: 5 mg of dimethyl sulfide Sigma Aldrich PA in 1 liter of pasteurized fresh milk cream “Fazenda Bela Vista^®^” brand
Oxidized oil flavor	The flavor of oxidized oil, known as rancid oil	None: Pasteurized fresh milk cream “Fazenda Bela Vista^®^” brand Moderate/Medium: 3 mg of butiric acid in 1 liter of pasteurized fresh milk cream “Fazenda Bela Vista^®^” brand Strong: 5 mg mg of butiric acid in 1 liter Pasteurized Fresh Milk Cream “*Fazenda Bela Vista*^®^” Brand
Sweetness	Characteristic taste of sucrose	None: 100 g plain yogurt *Salute^®^* brand + 4 g of sucrose Moderate/Medium: 100 g of plain yogurt Salute^®^ brand and 10 g of sucrose Strong: 100 g of plain yogurt Salute^®^ brand and 16 g of sucrose
Sourness	Acidic taste, also known as being sour, is present in dairy products such as yogurt, curd, and fermented milk	None: Pasteurized fresh milk cream “Fazenda Bela Vista” brand Moderate/Medium: 10 mg of lactic acid and 500 mL of pasteurized fresh milk cream “Fazenda Bela Vista^®^” brand Strong: 10 mg of lactic acid and 1000 mL of pasteurized fresh milk cream “Fazenda Bela Vista^®^” brand
Salty	Characteristic taste of salty foods such as cream cheese	None: Pasteurized fresh milk Verde Campo*(R)* Moderate/Medium: 50 mL of pasteurized fresh milk cream “Fazenda Bela Vista^®^” brand and Verde Campo^®^ and 0.05 g of cream cheese Vigor^®^ diluted in 10 mL of whole milk Shefa^®^ UHT Strong: 50 mL of pasteurized fresh milk cream “Fazenda Bela Vista^®^” brand and 0.2 g of cream cheese Vigor^®^ diluted in 10 mL of whole milk Shefa^®^ UHT
Bitterness	Characteristic taste of products containing caffeine	None: Whole milk Ninho^®^ UHT Moderate/Medium: 0.02 g of caffeine Sigma Aldrich PA in 100 mL of whole milk Ninho^®^ UHT Strong: 0.04 g of caffeine Sigma Aldrich PA in 100 mL of whole milk Ninho^®^ UHT
Astringency	A sensation of “tying” the mouth, such as green banana and cashew pulp	None: Whole milk Ninho UHT Moderate/Medium: 20 mL of whole milk Ninho^®^ UHT and 0.1 g of tannic acid Sigma Aldrich PA Strong: 20 mL of whole milk Ninho^®^ UHT and 0.5 g of tannic acid Sigma Aldrich PA
Vanilla flavor	Characteristic flavor of foods containing vanilla essence	None: Pasteurized fresh milk cream “Fazenda Bela Vista^®^” brand Moderate/Medium: 3 mL of vanilla essence IFF in 1000 mL of pasteurized fresh milk cream “Fazenda Bela Vista^®^” brand Strong: 10 mL of vanilla essence IFF in 1000 mL of pasteurized fresh milk cream “Fazenda Bela Vista^®^” brand
Flavor caramel	Characteristic flavor of sugar subjected to high temperatures, such as pudding syrups	None: Whole milk Ninho^®^ UHT Moderate/Medium: 20 grams of caramelized sugar dissolved in 1000 mL of whole milk Ninho^®^ UHT Strong: 50 g of inverted sucrose solved in 1000 mL of whole milk Ninho^®^ UHT
Sweet aftertaste	Sweet taste that lingers after swallowing the food	None: 1000 mL of whole milk Ninho^®^ UHT Moderate/Medium: 100 mL of whole milk Ninho^®^ UHT and 0.2 g of sodium saccharin Sigma Aldrich Strong: 100 mL of whole milk Ninho^®^ UHT and 0.5 g of sodium saccharin Sigma Aldrich
Bitter aftertaste	Bitter taste that lingers after swallowing the food	None: 1000 mL of whole milk Ninho^®^ UHT Moderate/Medium: 100 mL of Whole Milk *Ninho^®^* UHT and 2 g of stevia from Steviafarma^®^ Strong: 100 mL of whole milk Ninho^®^ UHT and 4 g of stevia from Steviafarma^®^
Milk cream flavor	Characteristic flavor of milk cream	None: Water Moderate/Medium: 100 mL of whole milk Ninho^®^ UHT and 50 g of fresh milk cream Nestle Strong: 100 mL of whole milk Ninho^®^ UHT and 100 g of fresh milk cream Nestle^®^
Butter flavor	Characteristic flavor of whipped cream	Slight/Weak: Freshly prepared cream from the surface formed on fresh whole milk by Fazenda Bella Vista^®^ after boiling and cooling, homogenized using a pistil in a mortar Moderate/Medium: 40 mg of diacetyl in 1 liter of fresh pasteurized cream by Fazenda Bella Vista^®^ brand Strong: 100 mg of diacetyl in 1 liter of fresh pasteurized cream by Fazenda Bella Vista^®^ brand
Creamy in the mouth	Creaminess in the mouth	Slight/Weak: 100 g in 1 liter of fresh pasteurized cream by Fazenda Bella Vista^®^ brand and 200 mL of whole milk Ninho^®^ UHT Moderate/Medium: 100 g of fresh pasteurized cream by Fazenda Bella Vista^®^ brand and 50 mL of whole milk Ninho^®^ UHT Strong: Chandelle^®^ dessert
Yogurt coating inside the mouth	Sensation of certain foods sticking to the tongue and palate	None: Deionized water at room temperature Moderate/Medium: 100 g of fresh milk cream Nestle^®^ and 100 mL of whole milk Ninho^®^ Strong: Fresh milk cream Nestle^®^ at room temperature
Mouth-filling	Characteristic of certain foods filling the mouth evenly and quickly	Slight/Weak: Cornstarch porridge prepared with 200 mL of milk, 10 g of cornstarch, 20 g of sugar, and 20 g of Nestlé^®^ cream without whey Moderate/Medium: Cornstarch porridge prepared with 200 mL of milk, 15 g of cornstarch, 25 g of sugar, and 20 g of Nestlé^®^ cream without whey Strong: Cornstarch porridge prepared with 200 mL of milk, 25 g of cornstarch, 25 g of sugar, 20 g of Nestlé^®^ cream without whey, and 100 g of melted Nestlé^®^ milk chocolate bar
Greasy	Slippery sensation between the tongue and palate when eating a high-fat food	None: Greek yogurt Danone^®^ light Moderate/Medium: 100 g of Greek yogurt Danone^®^ and 20 g of Nestlé^®^ cream without whey Strong: 100 g of Greek yogurt Danone and 70 g of Nestlé^®^ cream without whey
Aeration	Sensation of food containing air and lightness	None: Pasteurized fresh milk cream “Fazenda Bela Vista^®^” brand Moderate/Medium: Danone^®^ Greek yogurt mixed in a blender for 1 min. Strong: Danone Greek^®^ yogurt mixed in a blender for 3 min.
Viscous	Sensation of threads of creamy food forming between the tongue and the palate	None: Water Moderate/Medium: Vigor^®^ light cream cheese Strong: Orange blossom honey Melbee^®^ brand
Perception of lightness	Sensation of airy food, like whipped egg whites	Slight/Weak: Pasteurized fresh milk cream “Fazenda Bela Vista” brand Moderate/Medium: Egg whites freshly whipped in a planetary mixer, added with 10% cornstarch Strong: Egg white freshly whipped in a planetary mixer
Lump formation	Division that some creamy foods make with parts denser than others	None: Pasteurized fresh milk cream “Fazenda Bela Vista^®^” Moderate/Medium: 50 g of pasteurized fresh milk cream “Fazenda Bela Vista^®^” and 50 g of plain yogurt light Batavo^®^ Strong: 10 g of pasteurized fresh milk cream “Fazenda Bela Vista^®^” and 50 g of plain yogurt light Batavo^®^
Homogeneity in the mouth	Uniformity of food in the mouth	Slight/Weak: Plain yogurt light Batavo^®^ Moderate/Medium: Light cream cheese Danúbio^®^ Strong: Dessert Chandelle^®^
Texture in mouth	Property of some foods that cause a sensation of firmness (or the opposite, softness)	Slight/Weak: Pasteurized fresh milk cream “Fazenda Bela Vista^®^” Moderate/Medium: Danone^®^ plain yogurt Strong: Cheddar cheese Polenghi^®^

**Table 3 foods-14-00130-t003:** Means * of descriptor terms of Greek yogurt.

Descriptor Terms	P1	P2	P3	P4	P5	P6	P7	P8	1-GYMML	2-GYMML	3-GYNML
Yellow color	5.1 c	6.5 b	3.1 f	5.7 c	2.1 g	6.3 b	4.0 e	4.5 d	7.1 a	7.2 a	6.4 b
Apparent homogeneity	8.8 a	8.9 a	5.7 b	8.8 a	8.8 a	8.6 a	8.8 a	8.7 a	4.5 c	8.7 a	8.8 a
Syneresis	6.1 a	0.1 b	0.2 b	0.2 b	0.3 b	6.2 a	0.2 b	0.2 b	5.7 a	0.1 b	0.2 b
Texture when picking up with a spoon	8.3 a	8.4 a	8.1 a	7.6 b	7.9 a	8.3 a	8.2 a	8.4 a	4.2 c	7.2 b	8.5 a
Filament formation when picking up with a spoon	3.3 c	3.5 c	5.4 b	1.1 d	5.3 b	1.2 d	1.1 d	1.2 d	8.4 a	5.3 b	1.2 d
Shine	8.4 a	8.6 a	8.5 a	8.3 a	8.5 a	8.5 a	8.6 a	8.5 a	6.8 b	8.5 a	8.5 a
Surface uniformity	8.3 a	8.5 a	8.2 a	8.5 a	8.5 a	8.5 a	8.4 a	8.4 a	4.5 b	8.4 a	8.5 a
Acid aroma	3.4 d	4.4 c	5.1 b	3.3 d	3.5 d	3.6 d	3.5 d	3.4 d	6.7 a	3.5 d	3.6 d
Sweet aroma	7.7 a	7.6 a	7.8 a	4.2 b	7.6 a	7.7 a	7.5 a	4.1 b	4.0 b	7.6 a	7.7 a
Dairy aroma	8.3 a	8.2 a	8.2 a	5.4 b	8.0 a	5.7 b	8.1 a	3.7 c	3.9 c	8.1 a	8.3 a
Buttery aroma	4.6 c	7.5 b	4.4 c	3.0 d	7.6 b	4.8 c	7.6 b	7.7 b	2.8 d	4.7 c	8.2 a
Vanilla aroma	5.3 c	3.7 d	5.4 c	3.5 d	7.5 a	7.6 a	7.6 a	6.1 b	3.6 d	7.4 a	7.6 a
Diacetyl aroma	3.8 b	6.1 a	3.3 c	1.3 d	6.1 a	6.1 a	6.0 a	5.9 a	1.2 d	3.7 b	6.1 a
Caramel aroma	5.1 b	5.0 b	5.2 b	3.4 d	7.2 a	3.5 d	7.2 a	4.2 c	1.5 e	7.3 a	7.1 a
Cheese aroma	0.2 b	2.3 b	0.3 b	4.7 a	0.1 b	0.2 b	0.1 b	0.3 b	4.6 a	0.3 b	0.4 b
Dairy flavor (milk)	7.0 b	4.5 c	4.8 c	8.1 a	4.7 c	4.6 c	8.3 a	4.6 c	2.7 d	7.8 a	8.2 a
Metallic flavor	7.1 a	0.0 c	0.0 c	0.0 c	0.0 c	0.0 c	0.0 c	0.0 c	4.8 b	0.0 c	0.0 c
Cheese flavor (sour milk)	0.0 c	0.0 c	1.8 b	0.0 c	0.0 c	0.0 c	0.0 c	2.8 b	8.2 a	0.0 c	0.0 c
Cottage cheese flavor	0.3 c	0.2 c	0.4 c	0.4 c	0.4 c	0.2 c	0.3 c	0.4 c	5.5 a	3.1 b	0.3 c
Sulfurous flavor	0.0 d	0.0 d	5.2 b	0.0 d	4.8 c	0.0 d	0.0 d	0.0 d	5.7 a	0.0 d	0.0 d
Rancid oil flavor	7.5 a	0.0 d	7.7 a	0.0 d	4.7 b	0.0 d	0.0 d	0.0 d	2.8 c	0.0 f	0.0 f
Sweetness	6.6 b	2.3 e	7.3 a	6.5 b	6.6 b	4.5 c	7.3 a	3.8 d	6.5 b	7.4 a	7.2 a
Acidity	4.6 d	7.9 a	4.4 d	4.5 d	4.5 d	6.5 b	4.7 d	6.6 b	4.7 d	4.6 d	5.5 c
Salty	0.2 c	0.3 c	0.2 c	0.3 c	0.3 c	0.3 c	0.1 c	0.2 c	5.6 a	4.3 b	0.2 c
Bitterness	0.0 c	0.0 c	0.0 c	0.0 c	1.5 b	0.0 c	0.0 c	0.0 c	6.7 a	0.0 c	0.0 c
Astringency	2.2 c	2.7 b	2.6 b	3.1 a	3.0 a	3.2 a	2.4 b	3.0 a	3.2 a	2.3 b	2.5 b
Vanilla flavor	4.8 d	4.0 e	6.9 c	4.1 e	6.8 c	4.0 e	6.9 c	4.2 e	0.8 f	7.8 b	8.7 a
Caramel flavor	7.2 a	4.1 b	7.3 a	4.2 b	7.2 a	7.3 a	7.1 a	4.0 b	4.2 b	4.2 b	7.2 a
Residual sweetness	6.3 b	2.3 d	7.0 a	6.2 b	6.4 b	4.2 c	6.6 b	3.5 c,d	6.5 b	6.8 b	6.7 b
Residual bitterness	0.2 c	0.3 c	0.1 c	0.3 c	1.5 b	6.1 a	0.2 c	0.2 c	7.5 a	0.2 c	0.3 c
Milk cream flavor	6.7 b	4.2 c	4.0 c	7.8 a	4.7 c	4.6 c	7.9 a	4.5 c	2.9 d	7.3 a	7.8 a
Buttery flavor	3.4 c	6.4 b	3.6 c	2.2 d	6.5 b	3.9 c	6.5 b	6.5 b	2.0 d	3.8 c	7.3 a
Creamy in the mouth	8.7 a	8.8 a	8.8 a	8.7 a	8.8 a	8.6 a	8.9 a	8.7 a	2.8 b	7.8 a	8.6 a
Grip in the mouth	7.5 a	7.7 a	7.3 a	7.6 a	7.4 a	7.5 a	7.6 a	7.8 a	3.8 b	7.6 a	7.6 a
Mouth-filling	7.9 a	8.0 a	7.9 a	8.1 a	7.8 a	7.9 a	7.8 a	7.7 a	4.3 b	8.0 a	7.7 a
Greasy	2.3 b	2.0 b	2.2 b	2.3 b	2.4 b	6.2 a	2.1 b	2.1 b	2.2 b	2.2 b	2.4 b
Aeration	6.5 c	6.5 c	7.5 b	6.3 c	6.5 c	8.1 a	6.6 c	6.6 c	6.4 c	6.5 c	6.5 c
Viscous	4.4 b	4.7 b	4.3 b	4.5 b	4.6 b	4.7 b	4.5 b	4.7 b	7.8 a	4.5 b	4.5 b
Perception of lightness	7.0 b	7.0 b	7.9 a	7.2 b	7.0 b	8.0 a	7.1 b	6.9 b	7.1 b	6.9 b	7.1 b
Lump formation	0.6 b	0.6 b	0.4 b	0.6 b	0.7 b	0.5 b	0.5 b	0.7 b	2.1 a	0.6 b	0.6 b
Homogeneity in the mouth	7.9 a	8.2 a	8.0 a	8.2 a	8.0 a	8.0 a	8.2 a	7.8 a	6.7 b	8.1 a	8.1 a
Texture in the mouth	8.0 b	8.7 a	8.2 b	8.8 a	8.3 b	8.1 b	8.7 a	8.2 b	4.9 c	8.5 a	8.7 a

* Means with same letters in the same line do not differ significantly according to Tukey’s test (*p* > 0.05).

**Table 4 foods-14-00130-t004:** Demographic information of consumer participants of acceptance: gender, education level, age, and employment status.

Question	Response	Consumers Participants %
Gender	Female	54.7
	Male	43.3
	Others	3
Education Level	Associate degree	29
	Bachelor’s degree	54.0
	Master’s degree	11
	Doctorate	6
Age	18–25	25.0
	26–30	28
	31–35	35.0
	36–40	12.0
	41–45	7.0
Employment status	College student	18.0
	Unemployed	3.0
	University employed	32.0
	Self-employed	34.0
	Homemaker	13.00

**Table 5 foods-14-00130-t005:** Means * of solid soluble °Brix, pH, and instrumental texture (F/g).

Sample	°Brix	pH	Textura (F/g)
P1	20.63 a	4.27 a	27.651 c
P2	20.73 a	4.09 a	24.124 e
P3	20.43 a	4.27 a	28.648 b
P4	18.60 b	4.26 a	30.900 a
P5	20.57 a	4.36 a	28.170 b
P6	16.13 d	4.16 a	18.040 f
P7	18.00 b	4.24 a	25.149 d
P8	17.00 c	4.15 a	18.837 f
1-GYMML	20.20 a	4.36 a	14.468 g
2-GYMML	18.86 b	4.22 a	18.264 f
3-GYNML	18.30 b	4.17 a	21.384 e

* Means with same letters in the same column do not differ significantly according to Tukey’s test (*p* > 0.05).

## Data Availability

The original contributions presented in this study are included in the article. Further inquiries can be directed at the corresponding author.

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
