# Peer review of "Unlocking Consumer Preferences: Sensory Descriptors Driving Greek Yogurt Acceptance and Innovation"

_foods, 2025, doi:10.3390/foods14010130_

Round 1

Reviewer 1 Report

Comments and Suggestions for Authors

The article is a comprehensive analysis of the sensory attributes that influence consumer acceptance of Greek yogurt. The introduction to the manuscript provides basic information on the subject of the study, while the literature review could be enriched with more recent publications, especially those from the last 5 years. The study is well-structured and uses a solid methodology, including sensory evaluations, quantitative descriptive analysis, consumer acceptance tests and physicochemical analyses such as pH determination, soluble solids content and instrumental texture analysis; which allows for a comprehensive understanding of the factors influencing consumer preferences. It should also be emphasized that the methodology is in line with recognized standards (e.g. ISO 8589:2007), which increases the reliability and repeatability of the study. The manuscript effectively highlights the factors that should be considered when trying to fit into consumer preferences. The results are presented in a clear manner and supported by statistical analyses, which allows for drawing reliable conclusions. It should be noted that the reference list lacks DOI, which makes it difficult to find cited publications. It would be suggested to supplement this information and systematize the references in accordance with the journal requirements, which will definitely facilitate interested readers to further deepen their knowledge in the indicated area. The manuscript has the potential to make a significant contribution to understanding consumer preferences for Greek yogurt, offering insights that are valuable both scientifically and practically.

Author Response

Reviewer 1

Comments 1: ..."The introduction to the manuscript provides basic information on the subject of the study, while the literature review could be enriched with more recent publications, especially those from the last 5 years."...

Response 1:  We agree with the comment. More recent publications have been added and are highlighted in red in the “Introduction” section (lines 47, 68, 71, and 83). These references are also listed in the “References” section, numbered 9, 18, 19, and 21.

Comments 2: ..."It should also be emphasized that the methodology is in line with recognized standards, which increases the reliability and repeatability of the study...."

Response 2:  Thank you for your comments. The standard methodology has been included in Section 2.3.1, line 146 (highlighted in red, reference 27)

Comments 3: ..."It should be noted that the reference list lacks DOI, which makes it difficult to find cited publications."...

Response 3: We agree. Thank you for your comments. All available DOIs have been added to the references.

Reviewer 2 Report

Comments and Suggestions for Authors

The study is very well interesting, and it is comprehensive. 

I suggest the following:

1-      Include some actual values for significant findings in the abstract, for example add Tukey’s test and P values (P< 0.05) for “higher consumer acceptance was characterized by sensory attributes such as "high texture in the mouth…

2-      Add some more important nutritional attributes of Greek Yogurt such as probiotic property. 

3-      It would be helpful to add a demographic table that highlights the participants age, education, socioeconomic status, etc…

4-      Have the authors looked for correlation between some of the demographic data and sensory descriptors? And age and education?

5-      Table 1, suggest only 2 decimals points would be sufficient. 

6-      Summarize some of Table 3 findings of sensory attributes and actability. 

7-      It will be very interesting to look at the relationship (correlation) between purchasing intention and some of the demographic data. 

8-      What are the main factors in their buying decision? 

9-      Basically, what are the main factors, age, education, socioeconomic, etc. and acceptability and purchasing intention?

Including these suggestions/recommendations and careful editing may strengthen the quality of the manuscript. Otherwise, the paper is well written. 

Author Response

Thank you for your time to review this manuscript. Thank you for feedback on our paper entitled “Unlocking Consumer Preferences: Sensory Descriptors Driving Greek Yogurt Acceptance and Innovation” (foods-3377770).

We sincerely appreciate and value the insightful contributions, including their suggestions and correction requests. These have undoubtedly enhanced the quality and coherence of our manuscript.

Below, we present our responses to each comment and suggestion. The modifications made to the revised manuscript are highlighted in blue letters for ease to review.

Thank you for considering this manuscript.

Reviewer 3 Report

Comments and Suggestions for Authors

The paper is well constructed and the methodology was very well structured and applied. 

I would like to suggest the authors to write foreign names in italic and add the registered mark symbol under well known brands (such as Nestlé). Also, given that trained panelists were used, I think that authors should not use the word “consumer” on the manuscript and title. 

Although the acceptance test was well constructed, in order to reflect consumers preference, untrained panelists should be used. In this sense, I understand the determination of drivers of liking the main product of this study.

Please describe the number of panelists on the abstract as well. 

Author Response

Comments 1 - I would like to suggest the authors to write foreign names in italic and add the registered mark symbol under well known brands (such as Nestlé). Also, given that trained panelists were used, I think that authors should not use the word “consumer” on the manuscript and title. 

Response 1 - We agree. The information was added in all brands.

Comments 2 - Also, given that trained panelists were used, I think that authors should not use the word “consumer” on the manuscript and title. 

Response 2 - Dear Reviewer, in our research group, these methodologies are employed in accordance with the guidelines provided by leading experts in the field, as referenced in the following literature: • Meilgaard, M.; Civille, G.V.; Carr, B.T. Sensory Evaluation Techniques, 4th ed.; CRC Press: Boca Raton, FL, USA, 2006; pp. 25–38. • Stone, H.; Bleibaum, R.I.; Thomas, H.A. Sensory Evaluation Practices, 4th ed.; Academic Press, New York, 2012, 438 p. These two books emphasize that descriptive tests, such as Quantitative Descriptive Analysis (QDA), must be conducted with pre-selected and trained individuals, whose performance is statistically validated using metrics such as pFampstra and pFrepetition (as described in Sections 2.3 and 2.4 of the present manuscript). The same references highlight that consumer tests, aimed at determining product acceptance, preference, or purchase intention, should be carried out with regular consumers of the product. This approach was adopted and described in Section 2.5 of this study. These authors also emphasize the importance of multivariate analyses to identify characteristics that contribute positively or negatively to acceptance. Such analyses should be applied to datasets comprising descriptive quantitative results from trained panelists and acceptance data from frequent consumers of the product under study. In this study, we demonstrate which characteristics (as determined by QDA) drive consumer acceptance of the tested products. For this reason, we would like to maintain the title as it is currently stated. Thank you very much for your attention and consideration.

Comments 3 -Please describe the number of panelists on the abstract as well

Response 3 - We agree. The information was added in line 17-18 (green letters)
